# Distribution of KIR Genes and Their HLA Ligands in Different Viral Infectious Diseases: Frequency Study in Sicilian Population

**DOI:** 10.3390/ijms232415466

**Published:** 2022-12-07

**Authors:** Mattia Emanuela Ligotti, Anna Aiello, Giulia Accardi, Anna Calabrò, Marcello Ciaccio, Claudia Colomba, Danilo Di Bona, Bruna Lo Sasso, Fanny Pojero, Antonino Tuttolomondo, Calogero Caruso, Giuseppina Candore, Giovanni Duro

**Affiliations:** 1Laboratory of Immunopathology and Immunosenescence, Department of Biomedicine, Neurosciences and Advanced Diagnostics, University of Palermo, 90134 Palermo, Italy; 2Section of Clinical Biochemistry and Clinical Molecular Medicine, Department of Biomedicine, Neuroscience and Advanced Diagnostics, University of Palermo, 90127 Palermo, Italy; 3Department of Health Promotion, Maternal and Infant Care, Internal Medicine and Medical Specialties, University of Palermo, 90127 Palermo, Italy; 4Department of Emergency and Organ Transplantation, University of Bari Aldo Moro, 70124 Bari, Italy; 5Institute for Biomedical Research and Innovation, National Research Council of Italy, 90146 Palermo, Italy

**Keywords:** KIR gene, HLA, Sicilian population, COVID-19, HIV, HBV

## Abstract

Natural killer (NK) cells play a role in defence against viral infections by killing infected cells or by producing cytokines and interacting with adaptive immune cells. Killer immunoglobulin-like receptors (KIRs) regulate the activation of NK cells through their interaction with human leucocyte antigens (HLA). Ninety-six Sicilian patients positive to Human Immunodeficiency Virus-1 (HIV) and ninety-two Sicilian patients positive to SARS-CoV-2 were genotyped for KIRs and their HLA ligands. We also included fifty-six Sicilian patients with chronic hepatitis B (CHB) already recruited in our previous study. The aim of this study was to compare the distribution of KIR–HLA genes/groups of these three different infected populations with healthy Sicilian donors from the literature. We showed that the inhibitory *KIR3DL1* gene and the *KIR3DL1*/HLA-B Bw4 pairing were more prevalent in individual CHB. At the same time, the frequency of HLA-C2 was increased in CHB compared to other groups. In contrast, the HLA-C1 ligand seems to have no contribution to CHB progression whereas it was significantly higher in COVID-19 and HIV-positive than healthy controls. These results suggest that specific KIR–HLA combinations can predict the outcome/susceptibility of these viral infections and allows to plan successful customized therapeutic strategies.

## 1. Introduction

NK cells are important components of the innate immune system that play a key role in the host’s first line of defence against viral and tumor targets, providing immune surveillance and resistance to infection. NK cell cytotoxic and secretory functions are tightly regulated by the balance of activating and inhibitory signals from an arsenal of membrane receptors, including KIRs. They are characterized by two or three extracellular immunoglobulin-like domains (“2D” or “3D”) and a short (S) or long (L) intracytoplasmic tail. All inhibitory KIRs are characterized by a long intracytoplasmic tail while activating KIRs by a short one. Encoded by genes located on the long arm of chromosome 19 (19q13.4), within the leukocyte receptor complex, KIRs are highly polymorphic at the allelic and haplotypic level and interact with specific ligands represented by some class I HLA, expressed on their cell targets. To date, 15 functional KIR genes (*2DL1* to *2DL5A/B*, *3DL1* to *3DL3*, *2DS1* to *2DS5*, and *3DS1*) and 2 pseudogenes (*2DP1* and *3DP1*) have been described. Four of them are framework genes (*3DL2*, *3DL3*, *2DL4*, and *3DP1*) that are present with very few exceptions in all individuals [1]. However, numerous haplotypes with different gene content and allelic diversity segregate in human populations, creating considerable variability in the number of KIR genotypes observed in humans [2,3]. The number of officially named human KIR alleles is increasing and there are now over 1500 alleles that code for over 600 unique protein sequences [4]. 

Based on the number and type of genes encoding inhibitory and activating receptors, KIR haplotypes can broadly be classified as group A or B. The four framework KIR genes are found on almost all haplotypes, but the remaining haplotype gene content is variable, giving rise to genotype diversity. KIR genotype, defined as the repertoire of KIR genes present in an individual, can be classified as AA and Bx, where x can be A or B. Haplotype A is relatively conserved in gene content and is composed by the framework genes and a combination of *KIR2DL1*, *2DL3*, *3DL1*, *KIR2DS4*, and *KIR2DP1*. Group B haplotypes embrace all other KIR haplotypes with different combinations of activating and inhibitory KIR genes with one or more of the following genes: *KIR2DS1*, *2DS2*, *2DS3*, *2DS5*, *3DS1*, *2DL2* and *2DL5*. Thus, haplotype A has only an activating KIR, *KIR2DS4*, that commonly exists as alleles carrying a 22-bp deletion in exon 5 that results in a truncated non-functional protein [5]. Various alleles of *KIR2DS4* have been described, but few encode for cell-surface receptors [4]. Thus, individuals homozygous for KIR haplotype A with non-functional deletion variants of *KIR2DS4* do not have activating KIRs on the NK cell surface [6,7,8,9]. In a study on mother-to-infant HIV-1 transmission, the higher frequency of non-functional *KIR2DS4* variant was significantly associated with increased risk of in utero HIV-1 acquisition, particularly significant in infants homozygous for the haplotype A [10].

KIR ligands are HLA class I molecules that are expressed in all nucleated cells. The HLA genes map to the short arm of chromosome 6 (6p21.3) forming one of the most polymorphic gene clusters in the human genome. This genomic region is conventionally divided into three main sub-regions: class I, class II, and class III genes [11]. Among those of class I, HLA-A, -B and -C genes encoded the most important ligands for NK cells. The alleles of the HLA-C locus can be distinguished into two groups of ligands (C1 and C2) according to the amino acid present at position 80 of the α-1 domain of the α helix. HLA-C allotypes carrying asparagine (Ans) at position 80 (HLA-C1^Asn80^) are termed HLA-C group 1 (HLA-C1^Asn80^) and provide the ligand for KIR2DL2 and KIR2DL3. HLA C allotypes with lysine (Lys) at position 80 are called HLA-C group 2 (HLA-C2^Lys80^) and provide the ligand for KIR2DL1 [12]. Although KIR2DL2 and 2DL3 both bind HLA-C1^Asn80^ group, the KIR2DL3-HLA-C1 interaction appear to interact weaker than the 2DL2-HLA-C1 interaction. The activating KIR2DS1 had similar binding specificities as its inhibitory counterpart KIR2DL1, and it also binds HLA-C2, but at a lower affinity [13]. Since alleles among HLA-C1 and HLA-C2 are non-overlapping, individuals can be C1 homozygous (C1/C1), C2 homozygous (C2/C2) or C1/C2 heterozygous [8,12,14,15,16].

Among HLA-B alleles, only those bearing the HLA-Bw4 motif serve as ligands for KIRs. Like HLA-C, HLA-B Bw4 is divided into two groups based on the presence of isoleucine (I) or threonine (T) at positions 80 (HLA-B Bw4^I^ and HLA-B Bw4^T^, respectively), and this sequence dimorphism affects the interaction between HLA-Bw4 and KIR3DL1. HLA-B Bw4^I^ allotypes exhibit stronger inhibition through KIR3DL1 than HLA-B Bw4^T^ allotypes. In addition, the same epitope is also found on 4 HLA-A antigens (HLA-A*23/*24/*25/*32) and the ligand is called HLA-A Bw4 [8,14,15,16].

NK cell cytotoxic and cytokine-secreting activity are tightly controlled by these inhibitory and activating KIR–HLA interactions. The downregulation of HLA class I molecules in infected cells renders host cells potential targets for NK cells [12].

Given the role of KIRs in the immune response and their extensive genomic diversity, it is conceivable that KIR gene variation and KIR–HLA association affect resistance and susceptibility to the pathogenesis of many diseases, such as infectious diseases and autoimmune/inflammatory disorders, through modulation of NK activation, cytotoxicity and cytokine release [15].

Several lines of evidence suggest a strong contribution of KIR–HLA molecules to the outcome and susceptibility of different viral infections [17,18,19,20], including HIV, Hepatitis B virus (HBV), and SARS-CoV-2, although some data on the contribution of activating and inhibitory KIRs are in contrast, probably related to the different populations analysed. For example, a different KIR gene gradient between Northern and Southern Europe was observed, with a predominance of haplotype B carrying activating KIRs and strong inhibitory HLA ligands (HLA-C2 and HLA-B Bw4) in Southern Europe, while a lower frequency of activating KIR genes and predominance of weaker inhibitory ligands was observed in Northern Europe [21]. In Italy, on the other hand, a homogeneous frequency distribution of KIR genes but a different trend for HLA ligands was observed [22].

To gain insight into the mechanisms of resistance/susceptibility to viral infections, in the present paper we investigated the distribution of individual KIR genes, their HLA ligands, KIR–HLA associations and KIR haplotypes in three different cohorts belonging to the same geographical area (Sicily): (i) symptomatic and asymptomatic subjects positive to SARS-CoV-2 in absence of COVID-19 vaccination; (ii) HIV-infected individuals who were receiving antiretroviral therapy; (iii) HBV-positive subjects with CHB [18]. We also compared the distribution of KIR–HLA genes/groups of infected Sicilian patients with healthy Sicilian donors from the literature data [23], chosen as representative of the general Sicilian population, to assess if there are differences in KIR or HLA-ligand repertoire between our infected cohorts and healthy controls.

## 2. Results

To analyse the KIR repertoire and their HLA ligands in the Sicilian infected population, we first genotyped all subjects, counted which activating/inhibitory KIR and HLA class I genes are present in all individuals, and calculated their frequencies (Table 1). The purpose of Table 1 was only to present patient immunogenetic data. *KIR2DL1*, 3DL1 and 2DS4 genes, typical of haplotype A, were observed at a frequency greater than 90% in all groups. When distinguished from *KIR2DS4* full-length (the only functional activating KIR in haplotype A), *KIR2DS4* deletion variants were the most frequent gen (range %, 80–91). The most frequent functional activating KIR gene in all groups was *2DS2* (range 54–60%). Concerning HLA group frequency, our findings show a higher frequency of subjects homozygous for the HLA-C1 ligand group (C1/C1), and thus also of HLA-C1, in COVID-19 and HIV patients than in HBV (HLA C1 %, 75–84 vs. 64; HLA C1/C1 %, 30–33 vs. 13). The frequency of HLA-C2 and HLA-A Bw4 was higher in HBV individuals than in the other two groups (HLA-C2 %, 67–70 vs. 86; HLA-A Bw4 %, 24–32 vs. 57). Differences in frequencies of KIR–HLA group ligand interactions were also observed. The inhibitory *KIR3DL1*/HLA Bw4 association frequency, obtained from the sum of individual 3DL1 positivity in the presence of at least one allele with Bw4 epitope, was more frequent in HBV individuals (73 vs. 89%), with a greater contribution from the *KIR3DL1*/HLA-A Bw4 interaction (23–30 vs. 57%). The inhibitory *2DL1*/HLA-C2 interaction was more frequent in HBV individuals (82%) than in COVID-19 (65%) and HIV (39%) groups. Similarly, the combination of the *KIR2DS1* activating gene with its C2 ligand (2DS1/HLA-C2) was higher in HBV-positive individuals (41%) than in COVID-19 (21%) and HIV (16%). Although both *KIR2DL1* and *2DS1* bind HLA-C2 allele group, the affinity of these interactions may differ: *KIR2DS1*/HLA-C2 is probably a relatively weak interaction, while *KIR2DL1*/HLA-C2 may be relatively stronger [24].

Frequencies of haplotype AA in our cohorts were 20–26%, in line with that previously reported for the general Italian population and Sicilian healthy individuals, while the remaining individuals were Bx (AB + BB). A total of 39 distinct genotypes were observed (Figure 1a,b). Among these, 37 KIR genotypes are designed Bx, with a distribution variable within the three different groups. For example, haplotype Bx3 was more frequent in HBV (7.1%) than in COVID-19 (2.2%) and HIV (2.1%) individuals. The COVID-19 group showed the highest frequency of Bx5 (17%) among our groups (HIV, 9.4%; HBV, 11%). KIR AA1 was the most frequent in all COVID-19 (24%), HIV (23%), and HBV (20%) groups. Only two out of fifty-five AA1 individuals carried the *KIR2DS4* full-length gene. Another haplotype A (AA180) was found in COVID-19 and HIV cohorts at the same frequency (2%); all of them have deleted variants of *KIR2DS4*. Thus, almost all individuals carrying haplotype AA did not have functional KIR activators.

To evaluate differences in KIR and their HLA ligand genes between infected and healthy populations, we compared the frequencies of each KIR and HLA gene and KIR–HLA combination among our cohorts and healthy Sicilians. The results are shown in Table 2. The HLA-A Bw4 ligand was not tested in the healthy cohort and, thus, was excluded from the analysis. KIR inhibitor *2DL5A* was significantly higher in healthy subjects than in subjects with COVID-19 (*p* = 0.031) and HIV (*p* = 0.031). Regarding HLA ligands, a strong significant increase in the frequencies of HLA-C1 in COVID-19 (*p* = 0.0003) and HIV (*p* = 0.015) groups and of HLA-C2 in COVID-19 (*p* = 0.007), HIV (*p* = 0.021) and HBV (*p* < 0.0001) individuals was observed, which means a higher percentage of C1/C2 heterozygotes than the control population. Another noteworthy finding is the significant difference in the frequencies in the HLA-B alleles with Bw4 epitope, obtained from the individual presence of at least one HLA-B allele with Bw4 epitope, ligands for 3DL1 and 3DS1. Compared with healthy controls, a significantly higher frequency of HLA-B Bw4 was observed in COVID-19 (*p* < 0.0001), HIV (*p* = 0.0002) and HBV (*p* < 0.0001) cohorts.

The six most represented genotypes in Sicilian infected subjects were AA1, Bx2, Bx3, Bx4, Bx5, and Bx6. Their distributions are similar among healthy Sicilians, except for Bx2, higher in the control group, and Bx3 which was absent in the controls (Figure 2).

## 3. Discussion

Several lines of evidence relate KIR gene variation to the susceptibility and course of viral infections. Studies on KIR and HLA genotypes of individuals infected with HIV supported the role of the activating *KIR3DS1* in the presence or absence of its putative HLA-Bw4^I^ ligand, in a slower progression to acquired immune deficiency syndrome, lower mean viral load, and protection against opportunistic infections [25,26,27]. *KIR3DL1*, the inhibitory counterpart of *KIR3DS1*, is characterized by highly polymorphic alleles, leading to high variability in the expression. Co-expression of a *KIR3DL1* high-expression allotype with HLA-B Bw4^I^ induced slower disease progression and also a protective effect against HIV acquisition [28]. These results seem to contradict the model in which NK cell activation is protective. However, they can be explained by the importance of KIR–HLA class I interaction in establishing tolerance to healthy cells as well as in the activation potential of mature NK cells. During their development, NK cells must engage at least one inhibitory NK cell receptor with cognate HLA class I to achieve self-tolerance and become fully functional [29]. The stronger inhibitory interactions conferred by *KIR3DL1* during NK cell development can lead to a stronger NK cell reaction when the ligand is downregulated during viral infection. Similarly, the presence of *KIR2DL3* in combination with HLA-C1 was associated with lower viral load and disease progression as well as HIV resistance [27]. However, recent findings associated the co-carriage of *KIR2DL3* and HLA-C1 with higher viral load and increased mortality rates [30].

Genetic analyses of HBV-infected patients have demonstrated a highly protective role toward CHB of *KIR2DL3* in the presence of either HLA-A Bw4 or HLA-C2, but not in the presence of both [18]. In this study, *KIR2DL3*/HLA-C1 interaction has poor significance but is more represented in the resolved infections. Similarly, in a study on a Chinese cohort, the *KIR2DL1*-HLA-C2 interaction confers susceptibility to CHB, whereas *KIR2DL3* or *KIR2DL3* homozygote in the presence of HLA-C1/C1 genotype show protection against HBV persistence [31].

Studies on KIR–HLA interactions assume interest also to understand their role in NK cell defence against SARS-CoV-2 infection [20]. The comparison of KIR genes and KIR–HLA ligand combinations between Sardinian COVID-19 patients and Sardinian bone marrow donors have shown an overall increased frequency of KIR inhibitory genes in patients, with a statistically significant increase in *KIR2DL1* and *2DL3*, correlating them with increased susceptibility to SARS-CoV-2 infection. Among the COVID-19 group, those with the severe type of disease showed a statistically significant increase in the frequency of haplotype A/A as well as a significant reduction in *KIR2DS2* frequency compared to asymptomatic/paucisymptomatic patients [32]. Moreover, analysis of KIR expression showed a significant increase in 2DL1+ NK cells in severe compared to mild or moderate COVID-19 patients and SARS-CoV-2-uninfected controls [33].

Since NK cell activation is the result of a summation of activating and inhibitory signals, it is difficult to determine which KIRs influence the outcome (susceptibility) of infection without an analysis of haplotypes and HLA ligands to distinguish combined effects in different infected cohorts within the same geographical area. This kind of analysis provides a better understanding of the complex KIR involvement in innate immune responses to viral infections.

This study has potential limitations. First, the sample size is relatively small. Moreover, we did not detect any KIR–HLA gene differences between asymptomatic/paucisymptomatic and symptomatic COVID-19-positive individuals, therefore we included them in a single study cohort. Our work also has strengths. We studied individuals of the same geographical area, making it possible to compare the effects of a relatively conserved immunogenetic set-up between different viral infections.

In particular, the KIR–HLA profile of HBV-positive individuals seems to differ more than HIV and SARS-CoV-2, which instead show some similarities. Despite the small sample size, the inhibitory *KIR3DL1* gene was more prevalent in individual CHB (100%) than in COVID-19 (96%) and HIV-positive (92%) individuals and significantly higher than in healthy controls (*p* = 0.021, OR = ∞). Regarding KIR ligands, we found that HLA-C2 was more prevalent in individuals with HBV infection (86%) than in individuals with COVID-19 (70%) and HIV infection (80%). When compared to healthy controls, although statistically significant differences were observed with both COVID-19 (*p* = 0.007, OR = 2.68) and HIV-positive individuals (*p* = 0.021, OR = 2.34), the greatest difference was found with HBV-positive individuals (*p* < 0.0001, OR = 7.04), reinforcing the hypothesis that the HLA-C2 ligand may be involved in disease progression in these individuals. In contrast, the HLA-C1 ligand seems to have no contribution in HBV-positive individuals, whereas it was significantly higher in COVID-19 (*p* = 0.0003, OR = 4.37) and HIV (*p* = 0.015, OR = 2.55) compared to healthy controls.

Based on these results, there seems to be greater similarity in the KIR–HLA repertoire between SARS-CoV-2 and HIV than in HBV-infected individuals, both when compared to each other and healthy controls.

Therefore, further studies in homogeneous populations are mandatory and new knowledge about the mechanisms of control of infection could lead to better care as discussed below.

## 4. Materials and Methods

### 4.1. Studied Populations

Ninety-six Sicilian patients with HIV infection, responders to treatment with combined antiretroviral therapy (cART), and ninety-two SARS-CoV-2-positive Sicilian patients were consecutively recruited at the “Paolo Giaccone” Palermo University Hospital, Sicily, Italy. The determination of minimum sample sizes of HIV and SARS-CoV-2 cohorts could not be performed because no previous studies were available to estimate the expected prevalence of KIR–HLA in cohorts of Sicilian patients [34].

The Sicilian population is a Caucasian population originating from an ancient Mediterranean melting pot [35]. However, because immigration and intermarriage have been almost non-existent in the past century, the Sicilian Caucasian ethnicity of all participants was established if all four grandparents were born in Sicily. Accordingly, previous studies have shown that the distribution of HLA alleles and KIR genes in a sample of Sicilian population are in Hardy–Weinberg equilibrium [36,37], which confirms the non-existence of recent population mixtures.

HIV-positive individuals were recruited between 2019 and 2020. The protocol study was approved by the Ethics Committee of the “Paolo Giaccone” Palermo University Hospital (no. 08/2019). SARS-CoV-2-infected individuals were recruited between 2020 and 2021. The Ethics Committee of the “Paolo Giaccone” Palermo University Hospital approved the protocol study (no. 10/2020). All patients provided their written informed consent to participate, as well as for sampling and banking of the biological material. To respect privacy, all individuals were identified with an alphanumeric code. A database was created to organize the collected information.

The mean age of the ninety-six patients with HIV infection was 51.5 ± 11.8 years, with a prevalence of males (80.9%). Exclusion criteria were low cART adherence and the presence of major HIV drug resistance mutations. The mean age of the ninety-two COVID-19 patients was 62.3 ± 15.0 years, with a prevalence of males (60.9%). The SARS-CoV-2 infection was confirmed in all patients by RT-PCR from naso- and oropharyngeal swab, characterized by symptoms ranging from asymptomatic, i.e., presenting no or only mild symptoms and no pneumonia, to symptomatic with fever, respiratory symptoms and pneumonia, as confirmed by computed tomography. Participants who reported receiving COVID-19 vaccination before recruitment were excluded. The Ethics Committee of the “Paolo Giaccone” Palermo University Hospital approved the protocol study.

We also included fifty-six Sicilian patients with CHB at any stage of liver disease with or without any antiviral treatment already recruited in our previous study [18]. Moreover, we included the distribution of KIR genes and HLA groups of healthy Sicilian donors from the literature [23]. For the complete characteristics of these two populations, see the quoted papers.

### 4.2. Molecular Analysis

Peripheral whole blood samples were collected, and genomic DNA was extracted from leukocytes by Maxwell^®^ 16 Instrument (AS1000) using Maxwell^®^ 16 LEV Blood DNA Kit (Promega Corporation, Madison, WI 53711, USA). The KIR genotyping was performed by polymerase chain reaction sequence-specific primer technique for a total of 20 KIR genes using the KIR-TYPE kit (BAG Health Care GmbH, Lich, Germany): 2DL1, *2DL2*, *2DL3*, *2DL4*, *2DL5A*, *2DL5B*, *2DS1*, *2DS2*, *2DS3*, *2DS4-full*, *2DS4-del*, *2DS4-del(*008)*, *2DS5*, *3DL1*, *3DL2*, *3DL3*, *3DS1*, *2DP1*, *3DP1-full*, and *3DP1-del*. Five HLA-ligand groups including HLA-C1, HLA-C2, HLA-B Bw4^T^, HLA-B Bw4^I^, and HLA-A Bw4 were genotyped using the EPITOP-TYPE kit (BAG Health Care GmbH, Lich, Germany). All patients were stratified into two major groups according to homozygosity for KIR haplotype A (AA) and heterozygosity or homozygosity for KIR B haplotype (Bx, as AB + BB). KIR genotype profiles were assigned based on the presence/absence of each KIR of every individual in accordance with an online free tool/database the allelefrequencies.net [38].

### 4.3. Statistical Analysis

Crude comparisons of KIR gene frequencies, KIR haplotypes and HLA ligands between each infected group and Sicilian healthy controls were performed using 2 × 2 contingency tables, analysed by the Fisher’s exact test, and crude odds ratios were also calculated. Since no significant differences were found between symptomatic and asymptomatic SARS-CoV-2-positive subjects, data were pooled. The 4 framework genes (*3DL2*, *3DL3*, *2DL4*, and *3DP1*) were excluded from the statistical analysis because of their presence in all individuals. In addition, *2DS4-del (*008)* was excluded because it was not present in any individual. Statistical analyses were performed using GraphPad Prism 9.3.1 (GraphPad Software, San Diego, CA, USA). The threshold of significance was set at <0.05.

## 5. Conclusions

It is well known that activating KIR profiles are associated with decreased risk of some infectious disease outcomes, whereas inhibitory KIR genotypes with increased risk. Multiple factors complicate the interpretation of this association between KIR–HLA profiles and disease outcome, such as the limited understanding of KIR gene expression control and the extensive polymorphism of the KIR and HLA class I genes. However, it must be pointed out that, in outcome of viral infections, lifestyle (nutrition, physical activity, drug consumption, smoking), socio-economic parameters (such as level of education) and environment (exposure to xenobiotics and to multiple pathogens) should be considered since they mechanistically modify the immune response. In any case, since viral infections can have a profound effect on health and on quality of life, the identification of the mechanisms underlying the host immunogenetic control has important clinical implications. A more reliable result is obtained by comparing the immunogenetic profile of individuals affected by different types of viral infection belonging to the same population, to exclude interference from ethnic differences as much as possible and obtain a more comparable result. These types of studies are particularly important in driving efforts to reduce associated health complications in patients, in particular frail ones, and to potentially apply this knowledge clinically, i.e., in the development of new vaccines and in the identification of new therapeutic targets. It should also be noted that the present study is confined to Sicilian cohorts. Therefore, further studies and incremental experiments are necessary to define the role of KIR–HLA interaction in infectious diseases and to potentially apply this knowledge clinically. In fact, a huge comprehension of the impact of immunogenetic variants on immune mechanisms would offer the chance to identify those molecular pathways that are affected in the setting of the immune response against viral infections, leading to a more severe disease and to a more difficult resolution of the infections.

## Figures and Tables

**Figure 1 ijms-23-15466-f001:**
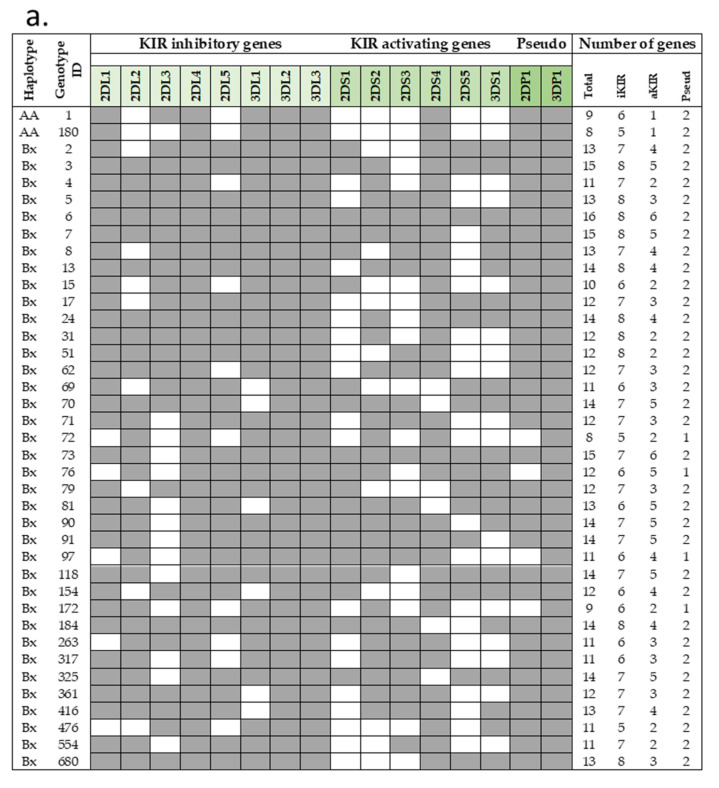
Genotype frequencies in infected Sicilian cohorts. (**a**) KIR genotype diversity in our infected cohorts. The presence or absence of a KIR gene is indicated by grey and white boxes, respectively. (**b**) Carrier frequencies of the KIR genotype in each cohort were determined as their percentages of the total number of individuals. Genotype ID refers to genotype classification according to allelefrequencies.net. aKIR, activating KIR; Del, delete variant of *KIR2DS4*; Full, full-length variant of *KIR2DS4*; iKIR, inhibitory KIR; Pseudo, pseudogene.

**Figure 2 ijms-23-15466-f002:**
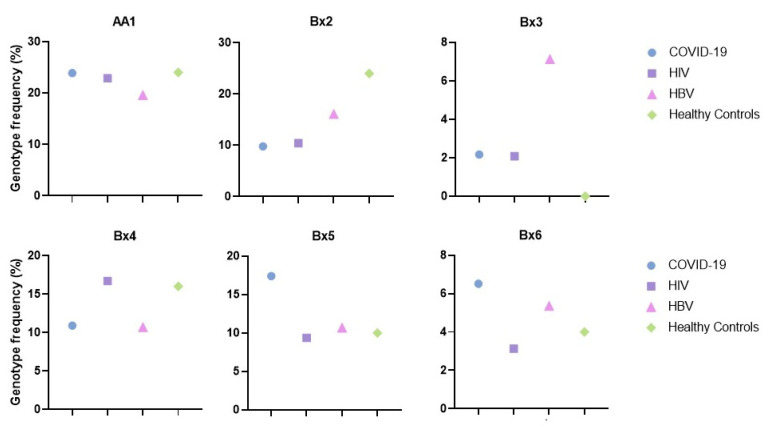
Frequencies of KIR genotypes –AA1, Bx2, Bx3, Bx4, Bx5, and Bx6 in infected and healthy control groups.

**Table 1 ijms-23-15466-t001:** KIR and HLA gene and haplotype frequencies (%) in different groups of infected populations.

	COVID-19 N_tot_ = 92 N (%)	HIV N_tot_ = 96 N (%)	HBV N_tot_ = 56 N (%)
**Haplotypes**			
AA	24 (*26*)	24 (*25*)	11 (*20*)
AB + BB	68 (*74*)	72 (*75*)	45 (*80*)
**KIR activating genes**			
*2DS1*	30 (*33*)	33 (*34*)	25 (*45*)
*2DS2*	50 (*54*)	58 (*60*)	33 (*59*)
*2DS3*	38 (*41*)	34 (*35*)	18 (*32*)
*2DS4 Full*	13 (*14*)	10 (*10*)	4 (*7*)
*2DS4 Del*	74 (*80*)	79 (*82*)	51 (*91*)
*2DS5*	25 (*21*)	27 (*28*)	21 (*38*)
*3DS1*	32 (*35*)	31 (*32*)	26 (*46*)
**KIR inhibitory genes**			
*2DL1*	88 (*96*)	89 (*93*)	53 (*95*)
*2DL2*	55 (*60*)	58 (*60*)	33 (*59*)
*2DL3*	79 (*86*)	73 (*76*)	47 (*84*)
*2DL5A*	30 (*33*)	31 (*32*)	26 (*46*)
*2DL5B*	32 (*35*)	30 (*31*)	18 (*32*)
*3DL1*	88 (*96*)	88 (*92*)	56 (*100*)
**HLA groups**			
HLA-C1	77 (*84*)	75 (*75*)	36 (*64*)
HLA-C2	64 (*70*)	64 (*67*)	48 (*86*)
HLA Bw4	71 (*77*)	77 (*80*)	50 (*89*)
HLA-B Bw4^T^	20 (*22*)	23 (*24*)	14 (*25*)
HLA-B Bw4^I^	50 (*54*)	52 (*54*)	31 (*55*)
HLA-A Bw4	22 (*24*)	41 (*32*)	32 (*57*)
HLA-C1/HLA-C1	28 (*30*)	32 (*33*)	7 (*13*)
HLA-C2/HLA-C2	15 (*16*)	24(*25*)	19 (*34*)
HLA-C1/HLA-C2	49 (*53*)	40 (*42*)	29 (*52*)
**KIR–HLA combinations**			
*3DL1*/HLA Bw4	67 (*73*)	70 (*73*)	50 (*89*)
*3DL1*/HLA-B Bw4^I^	46 (*50*)	44 (*46*)	31 (*55*)
*3DL1*/HLA-B Bw4^T^	18 (*20*)	21 (*22*)	14 (*25*)
*3DL1*/HLA-A Bw4	21 (*23*)	29 (*30*)	32 (*57*)
*3DS1*/HLA Bw4	24 (*26*)	23 (*24*)	18 (*32*)
*3DS1*/HLA-B Bw4^I^	20 (*22*)	16 (*80*)	12 (*21*)
*3DS1*/HLA-A Bw4	8 (*9*)	10 (*10*)	13 (*23*)
*2DL1*/HLA-C2	60 (*65*)	57 (*39*)	46 (*82*)
*2DL2*/HLA-C1	43 (*47*)	39 (*41*)	20 (*36*)
*2DL3*/HLA-C1	64 (*70*)	49 (*51*)	30 (*54*)
*2DS1*/HLA-C2	60 (*21*)	15 (*16*)	23 (*41*)
*2DS2*/HLA-C1	38 (*41*)	39 (*41*)	20 (*36*)

Percentages are shown in italics. COVID-19, Coronavirus disease 2019; HBV, Hepatitis B virus; HIV, Human Immunodeficiency virus; HLA, human leukocyte antigen; I, isoleucine; KIR, killer-cell immunoglobulin-like receptors; T, threonine.

**Table 2 ijms-23-15466-t002:** Differences in KIR genes and HLA ligands in infected Sicilian patients and healthy controls from Sicily.

	HC N_tot_ = 50 N (%)	COVID-19 vs. HC Crude-OR	*p*-Value	HIV vs. HC Crude-OR	*p*-Value	HBV vs. HC Crude-OR	*p*-Value
**KIR activating genes**							
*2DS1*	22 (*44*)	1.62	ns	0.77	ns	1.03	ns
*2DS2*	27 (*54*)	1.01	ns	1.30	ns	1.22	ns
*2DS3*	22 (*44*)	0.89	ns	0.70	ns	0.60	ns
*2DS4*	46 (*92*)	1.51	ns	1.10	ns	4.78	ns
*2DS5*	14 (*28*)	0.96	ns	1.0	ns	1.54	ns
*3DS1*	20 (*40*)	0.80	ns	0.71	ns	1.30	ns
**KIR inhibitory genes**							
*2DL1*	50 (*100*)	0.00	ns	0.00	ns	0.00	ns
*2DL2*	30 (*60*)	0.99	ns	1.02	ns	0.96	ns
*2DL3*	44 (*88*)	0.83	ns	0.43	ns	0.71	ns
*2DL5A*	26 (*52*)	0.44	**0.031**	0.44	**0.031**	0.80	ns
*2DL5B*	21 (*42*)	0.74	ns	0.63	ns	0.65	ns
*3DL1*	45 (*90*)	2.44	ns	1.22	ns	∞	**0.021**
**HLA groups**							
HLA-C1	27 (*54*)	4.37	**0.0003**	2.55	**0.015**	1.53	ns
HLA-C2	23 (*46*)	2.68	**0.007**	2.34	**0.021**	7.04	**<0.0001**
HLA-B Bw4	17 (*34*)	4.43	**<0.0001**	3.88	**0.0002**	10.14	**<0.0001**

HC, healthy controls; ns, not significant; OR, odds ratio; ∞, infinity. Note: The numbers in the boxes of column HS are the absolute number obtained from observed frequencies (%) of KIR genes and HLA ligands reported by Capittini et al., 2018 [23]. 2DS4 frequencies were calculated as the sum of the full and deleted variants. HLA-B Bw4 group frequencies were compared as the sum of HLA-B Bw4^I^ and HLA-B Bw4^T^. A *p*-value below 0.05 was considered statistically significant. *p*-values are shown in bold, while percentages are in italic. No significant differences were obtained by analysing frequencies according to age and sex.

## Data Availability

The data that support the findings of this study are available from the corresponding author, upon reasonable request.

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
