# Peer review of "Distribution of KIR Genes and Their HLA Ligands in Different Viral Infectious Diseases: Frequency Study in Sicilian Population"

_ijms, 2022, doi:10.3390/ijms232415466_

Round 1
Reviewer 1 Report
The authors aimed to compare the distribution of KIR/HLA genes/alleles of three different viral infected populations; however, several points need to be resolved before consideration for publications.
- Lines (80-97) are poorly written. Please replace them with better ones.
- Study populations are very small. How did you determine the Sample size? Or authors randomly chose the sample size? Or any other methods?
3. The authors measured the KIR and HLA gene and haplotype frequencies (%) in different groups of infected populations (table 1); however, there is no control that measured KIR and HLA gene and haplotype frequencies in patients with no viral infections in table 1.
4. There is no difference in the above parameters depending on the sex, or ages
- In Tables 1 and 3, it is difficult to identify which one is the number and which one is the percentage. Please make a difference between them.
- Figure 1 is too small to recognize. Please revise it.
- I can see table 3 in the text but not table 2. Please resolve the matter.
- Please improve the conclusion section and introduction.
Reviewer 2 Report
Pleas address the followings:
Major issues
The authors have to show that their cohorts are ethnically homogeneous. People living in a geographical region may have genetically heterogeneous.
Why some KIR could not be genotyped, because they were not detectable by the kit?
Table 1; line 118: OR?
Table 3 should also list frequencies of the listed KIR and HLAs, for the sake of a quick comparison.
Round 2
Reviewer 1 Report
Please see the attachment.

Author Response
Please, see the attachment.

Reviewer 2 Report
The authors need to verify ethnic origin of the study participants through genetic markers. There are concerns about using KIR gene frequencies from the literature.
Table 1 has OD in the legend. The table does not mention any OD.
Table 3. Same suggestion as in the previous review.
Author Response
Please, see the attachment.

Round 3
Reviewer 1 Report
A. I could not understand what the authors tried to make me understand about the way of sample size determination. The authors are requested to follow the following article to determine the minimum sample size if possible: “How to Calculate Sample Size for Different Study Designs in Medical Research?” 10.4103/0253-7176.116232
B. There is no difference in the above parameters depending on the sex, or ages. Regarding these comments authors replied that they did these two assays but the differences were insignificant. I think authors can mention these findings in the text.
Author Response
Please, see the attachment.

Reviewer 2 Report
The manuscript is much improved now. All my concerns have been addressed.
